# The kinetics of humoral response and its relationship with the disease severity in COVID-19

Lili Ren[1,2,12], Lulu Zhang [1,2,12], De Chang [3,12], Junwen Wang[4,12], Yongfeng Hu[5,12], Hong Chen[6,12], Li Guo[1,2], Chao Wu[1], Conghui Wang[1], Yingying Wang[1], Ying Wang[1], Geng Wang[1], Siyuan Yang [7], Charles S. Dela Cruz[8], Lokesh Sharma [8,13✉], Linghang Wang [9,13✉], Dingyu Zhang [10,11,13✉] & Jianwei Wang [1,2,13✉]

Coronavirus Disease 2019 (COVID-19) has caused a global pandemic. Here we profiled the humoral response against Severe Acute Respiratory Syndrome Coronavirus 2 (SARS-CoV-2) by measuring immunoglobulin (Ig) A, IgM, and IgG against nucleocapsid and spike proteins, along with IgM and IgG antibodies against receptor-binding domain (RBD) of the spike protein and total neutralizing antibodies (NAbs). We tested 279 plasma samples collected from 176 COVID-19 patients who presented and enrolled at different stages of their disease. Plasma dilutions were optimized and based on the data, a single dilution of plasma was used. The mean absorbance at 450 nm was measured for Ig levels and NAbs were measured using geometric mean titers. We demonstrate that more severe cases have a late-onset in the humoral response compared to mild/moderate infections. All the antibody titers continue to rise in patients with COVID-19 over the disease course. However, these levels are mostly unrelated to disease severity. The appearance time and titers of NAbs showed a significant positive correlation to the antibodies against spike protein. Our results suggest the late onset of antibody response as a risk factor for disease severity, however, there is a limited role of antibody titers in predicting disease severity of COVID-19.

[1] NHC Key Laboratory of Systems Biology of Pathogens and Christophe Mérieux Laboratory, Institute of Pathogen Biology, Chinese Academy of Medical Sciences & Peking Union Medical College, Beijing 100730, P.R. China. [2] Key Laboratory of Respiratory Disease Pathogenomics, Chinese Academy of Medical Sciences and Peking Union Medical College, Beijing 100730, P.R. China. [3] Third Medical Center of Chinese PLA General Hospital, Beijing 100039, P.R. China. [4] Wuhan 4th hospital, Wuhan 430023, P.R. China. [5] NHC Key Laboratory of Systems Biology of Pathogens, Institute of Pathogen Biology, Chinese Academy of Medical Sciences & Peking Union Medical College, Beijing 100730, P.R. China. [6] The Second Affiliated Hospital of Harbin Medical University, Harbin 150086, P.R. China. [7] Laboratory of Infectious Diseases Center of Beijing Ditan Hospital, Capital Medical University, Beijing 100015, P.R. China. [8] Section of Pulmonary and Critical Care and Sleep Medicine, Department of Medicine, Yale University School of Medicine, New Haven, CT 06520, USA. [9] Emergency Department of Infectious Diseases of Beijing Ditan Hospital, Capital Medical University, Beijing 100015, P.R. China. [10] Research Center for Translational Medicine, Wuhan Jin Yin-tan Hospital, Wuhan 430023, P.R. China. [11] Wuhan Research Center for Infectious Disease Diagnosis and Treatment, Chinese Academy of Medical Sciences, Wuhan 430023, P.R. China. [12] These authors contributed equally: Lili Ren, Lulu Zhang, De Chang, Junwen Wang, Yongfeng Hu, Hong Chen. [13] These authors jointly supervised this work: Lokesh Sharma, Linghang Wang, Dingyu Zhang, Jianwei Wang. ✉email: lokeshkumar.sharma@yale.edu; linghang.wang@ccmu.edu.cn; 1813886398@qq.com; wangjw28@163.com

Coronavirus disease 2019 (COVID-19) caused by Severe Acute Respiratory Syndrome Coronavirus 2 (SARS-CoV-2) emerged as a major pandemic that has spread across the globe with enormous healthcare and economic costs[1,2]. Humoral responses have been the major indicators of the disease severity during other viral infections in the lung including infections with SARS-CoV and influenza virus[3–5]. The SARS-CoV-2 genome encodes four structural proteins including spike (S), nucleocapsid (N), envelope, and membrane proteins[1]. The S and N proteins are the two major antigens in coronavirus that induce immunoglobulin (Ig) production[6]. Antibodies against the N protein are often induced in relative higher abundance than others, which are the main target for serological diagnosis[6,7]. The receptor-binding domains (RBD) present in the S1 region of S protein are the main target of neutralizing antibodies (NAbs) and can be a potential target for vaccine development[6,8,9]. The presence of NAbs is one of the most important indicators of clinical outcome and vaccination effectiveness during other respiratory viral infections[9,10].

Previously, we described the diagnostic value of SARS-CoV-2 antibody tests in the detection of COVID-19 in asymptomatic and possibly convalescent patients[7]. However, the prognostic value of the humoral responses has not been described in a systemic manner against various viral proteins of SARS-CoV-2. Further, the relationship between humoral responses and disease severity in COVID-19 remains to be known. In this study, we evaluate the disease onset time, positive rate, and titers of NAbs, IgA, IgM, and IgG against N and S protein, as well as IgM and IgG against RBD domains of SARS-CoV-2. Further, the correlations between the onset time and titers to the disease severity were investigated.

## Results

### The demographical and clinical information of the COVID-19 patients.
A total of 176 COVID-19 inpatients were recruited from three independent cohorts, in which 113 (64.2%) were males. The patients were aged from 18 to 82 years (mean of 49.0). The majority of the patients had clinical symptoms included fever (151, 85.8%), cough (136, 77.3%), and dyspnea (56, 31.8%) upon admissions (Supplementary Table 1). Comorbidities in the form of underlying diseases were recorded in 83 (47.1%) patients, including hypertension, diabetes, chronic respiratory diseases, coronary heart disease, stroke, etc.

The disease severity of the patients was classified as mild/moderate (79.5%, 140/176) or severe infections (20.5%, 36/176) which included ten (5.7%) deaths. Compared to mild/moderate (termed as mild hereafter) cases, the disease symptoms including fatigue (20.7% vs 41.7%) ($P = 0.010$), headache (12.1% vs 38.9%) ($P < 0.001$), muscle pain (14.3% vs 47.2%) ($P < 0.001$), sore throat (6.4% vs 41.7%) ($P < 0.001$) and dyspnea (22.1% vs 69.4%) ($P < 0.001$) were present at a higher frequency in severe cases ($\chi^2$ test) (Supplementary Table 1). The laboratory tests showed that the albumin decreased ($P = 0.001$) while the globulin increased ($P = 0.013$) significantly in severe patients (unpaired $t$-test). The leukocytes ($P < 0.001$) and neutrophil ($P < 0.001$) counts were higher, while lymphocytes were lower ($P = 0.009$) in severe cases compared to mild/moderate cases (unpaired $t$-test) (Supplementary Table 2).

### Characteristics of humoral antibodies in COVID-19 patients.
A total of 279 plasma samples were collected from the 176 patients, in which 103 patients provided two plasma samples (the interval between these two samples is 4 days), and 73 patients provided one sample. All the plasma samples were obtained between 1 and 46 days after symptom onset, with 60 samples between days 1 and 7, 108 samples between days 8 and 14, 104 samples between days 15 and 28, and 7 samples between days 29 and 46. First, the appearance time of IgA, IgM, IgG antibodies against N, S proteins, and the IgM and IgG antibodies against RBD protein, as well as the presence of NAbs against SARS-CoV-2 were examined. The presence of NAbs against SARS-CoV-2 was tested by microneutralization assay and further confirmed by using immunofluorescence assay and viral nucleic acid quantification by infecting the plasma-treated viruses on Vero cells using a qPCR method (Supplementary Fig. 1). All the antibodies were found to be positive as early as the first-day post symptom onset (PSO). The dynamic positive rate of serum antibodies was depicted in Fig. 1a and Supplementary files (Supplementary Fig. 2; Supplementary Data 1 and 2).

The specific antibodies of IgA, IgM, and IgG antibodies against N proteins were tested in 243 (87.1%), 216 (77.4%), and 207 (74.2%) samples, respectively. The N-IgA appeared at a more than 90% positive rate and remained above 80% in the first 3 weeks following a decrease to 60% at >21 days PSO. The positive rate of N-IgM increased to 80% on day 17 PSO reaching the plateau. N-IgG had a late surge and reached levels higher than other antibodies after day 23 PSO (Fig. 1a; Supplementary Table 3; Supplementary Data 1 and 2). The IgA, IgM, and IgG antibodies against S proteins were found in 119 (42.7%), 87 (31.2%), and 162 (58.1%) samples, respectively. The positive rates of S-IgG and IgA increased and remained high across the disease course, while the positive rate of S-IgM remained low across the disease course (Fig. 1a; Supplementary Table 3; Supplementary Data 1 and 2). The IgM, and IgG antibodies against RBD proteins were found in 152 (54.5%), and 158 (56.6%) samples, respectively. The positive rates of RBD-IgG increased over time, while the positive rate of RBD-IgM remained below 60% across the disease course (Supplementary Fig. 2; Supplementary Table 3; Supplementary Data 1 and 2).

### Antibody titers in patients with different disease severity in COVID-19.
The titers of all the antibodies increased during the disease course. N-IgG levels increased significantly between weeks 1–2 ($P < 0.001$) and 2–3 ($P = 0.010$) (unpaired $t$-test). N-IgA and IgM levels showed increased only between week 1 and 2 ($P < 0.001$). S-IgG levels increased continuously during three weeks PSO ($P = 0.015$; $P < 0.001$; $P = 0.007$, respectively), while S-IgA levels increased between weeks 1–2 ($P < 0.001$) and 2–3 ($P = 0.004$) (unpaired $t$-test). NAbs titers significantly increased between weeks 1–2 and 2–3 and reached an apparent plateau at 3 weeks PSO ($P = 0.001$, unpaired $t$-test) (Fig. 1b; Supplementary Data 2). The tabulated data of antibody positivity, appearance time, and levels were provided in the Supplementary file (Supplementary Table 3).

We then sought to evaluate whether the duration of appearance or the titers is related to the disease severity. We observed different kinetics of antibody appearance between mild/moderate and severe disease groups, with a significantly delayed onset of IgA ($P = 0.005$; $P = 0.008$), IgM ($P < 0.001$; $P = 0.009$), and IgG ($P < 0.001$; $P = 0.001$) against N and S proteins, respectively, as well as IgM ($P = 0.037$) and IgG ($P = 0.046$) against RBD in severe patients (unpaired $t$-test) (Fig. 2a; Supplementary Data 2). However, the antibody levels remained largely similar between mild/moderate and severe groups, except that S-IgA ($P = 0.034$) was higher in the third week, while RBD-IgM ($P = 0.030$) was lower in the fourth weeks, in severe cases (unpaired $t$-test) (Fig. 2b; Supplementary Data 2). NAbs increased significantly in mild/moderate cases in the third week (geometric mean titers, GMT 20.21) than that of the second week (GMT 10.64) ($P = 0.007$, unpaired $t$-test). There are no significant increments of NAbs titers in severe cases during the disease

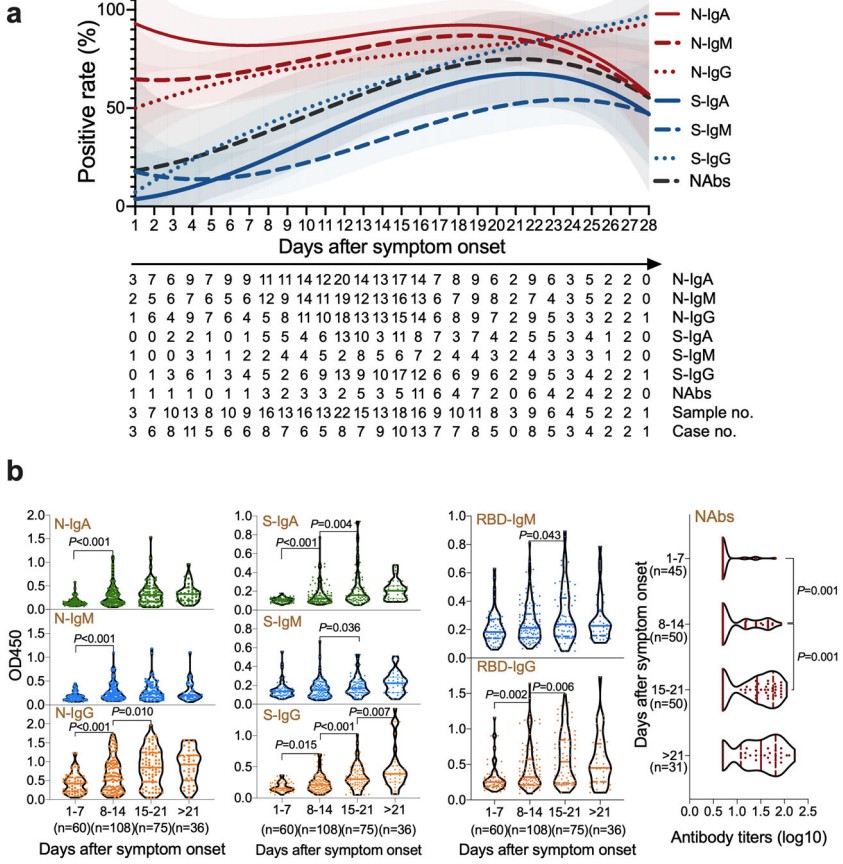

**Fig. 1 The characteristics of humoral responses in COVID-19. a** The positive rate of IgA, IgM, and IgG antibodies against SARS-CoV-2 nucleocapsid (N), spike (S) proteins and neutralizing antibodies (NAbs) in all the cases between 1 and 28 days after symptom onset. The fitted curve lines are created by Fit Spline program of Graphpad software where shadows indicating 95% confidence intervals. The table below the figure denotes the number of samples tested at each time point given in days and the positive rate at each day were used to draw the fitted curve. **b** Levels of IgM, IgA, and IgG antibodies and titers of NAbs against SARS-CoV-2 in plasma samples over course of the disease. Antibody titers were expressed as optical density (OD) value and NAbs titers are shown as geometric mean titers (GMTs). The median and quartiles were represented in the violin plots.

course (unpaired *t*-test) (Fig. 2b; Supplementary Data 2). Moreover, the positive rates of antibodies against SARS-CoV-2 in patients with different severity are also similar during the disease course (Supplementary Fig. 3; Supplementary Data 2).

**Correlations of neutralizing antibodies with binding-protein antibodies and disease severity in COVID-19 patients.** The S and RBD region are the key targets for vaccine design[9]. We then defined the associations of anti-S and -RBD antibodies with NAbs in reference to the clinical disease severity. The appearance time of the anti-S IgA, IgM and IgG ($r = 0.996$, 0.993, 0.993, respectively, $P < 0.001$) and anti-RBD IgM, IgG ($r = 0.996$, $r = 0.994$, $P < 0.001$) showed a strong positive correlation with the NAbs in all the patients (Spearman's rank correlation test) (Fig. 3a; Supplementary Data 2). The titers of IgA, IgM, IgG antibodies against S indicated as OD values were positively correlated with NAbs titers both in mild/moderate and severe cases. However, the titers of IgM, IgG antibodies against RBD protein showed positive correlations with NAbs titers only in mild/moderate cases, but not in severe cases (Fig. 3b; Supplementary Data 2). To define the time kinetics of the correlation, we found that the NAbs titers showed positive correlations with the IgA ($P = 0.010$) and IgM ($P = 0.002$) antibodies against S protein in the second week, with IgA ($P < 0.001$), and IgG ($P = 0.001$) antibodies against S protein and IgM ($P = 0.028$) antibodies against RBD protein in the third week (Spearman's rank correlation test) (Fig. 3c; Supplementary Data 2).

**Discussion**

COVID-19 emerged as a major healthcare challenge globally[1]. The disease spectrum varies widely, ranging from asymptomatic or very mild symptoms to severe disease and death. The major challenge in the disease management remains to predict the severe cases in a timely manner to provide aggressive intervention to limit the severe disease and subsequent mortality. Humoral response to viral infection is well-conserved mechanisms and the natural course of the antibody production can guide for possible vaccination strategies. We have previously shown that early humoral response can provide better diagnostic value if performed after 6 days post symptom onset[7]. Here we show delayed antibody responses in severe COVID-19 patients. The delayed response in severe patients was also observed in severe Middle East Respiratory Syndrome CoV (MERS-CoV) -infected patients, indicating similar humoral responses in emerging CoVs[11]. NAbs are the critical indicator to evaluate clinical outcomes and vaccination effects according to previous experience with SARS-CoV[3,10]. We showed the production of NAbs positively correlated with antibodies against S and RBD domains of SARS-CoV-2. However, surprisingly, we found that the NAbs titers showed no difference between mild/moderate and severe cases. This is different with that of SARS-CoV infections, where the NAb levels against SARS-CoV positively correlated with the clinical severity[3]. These data provided us with a possibility of whether differential antibody response can provide predictive value for the disease severity. Our data show that combining various antibody

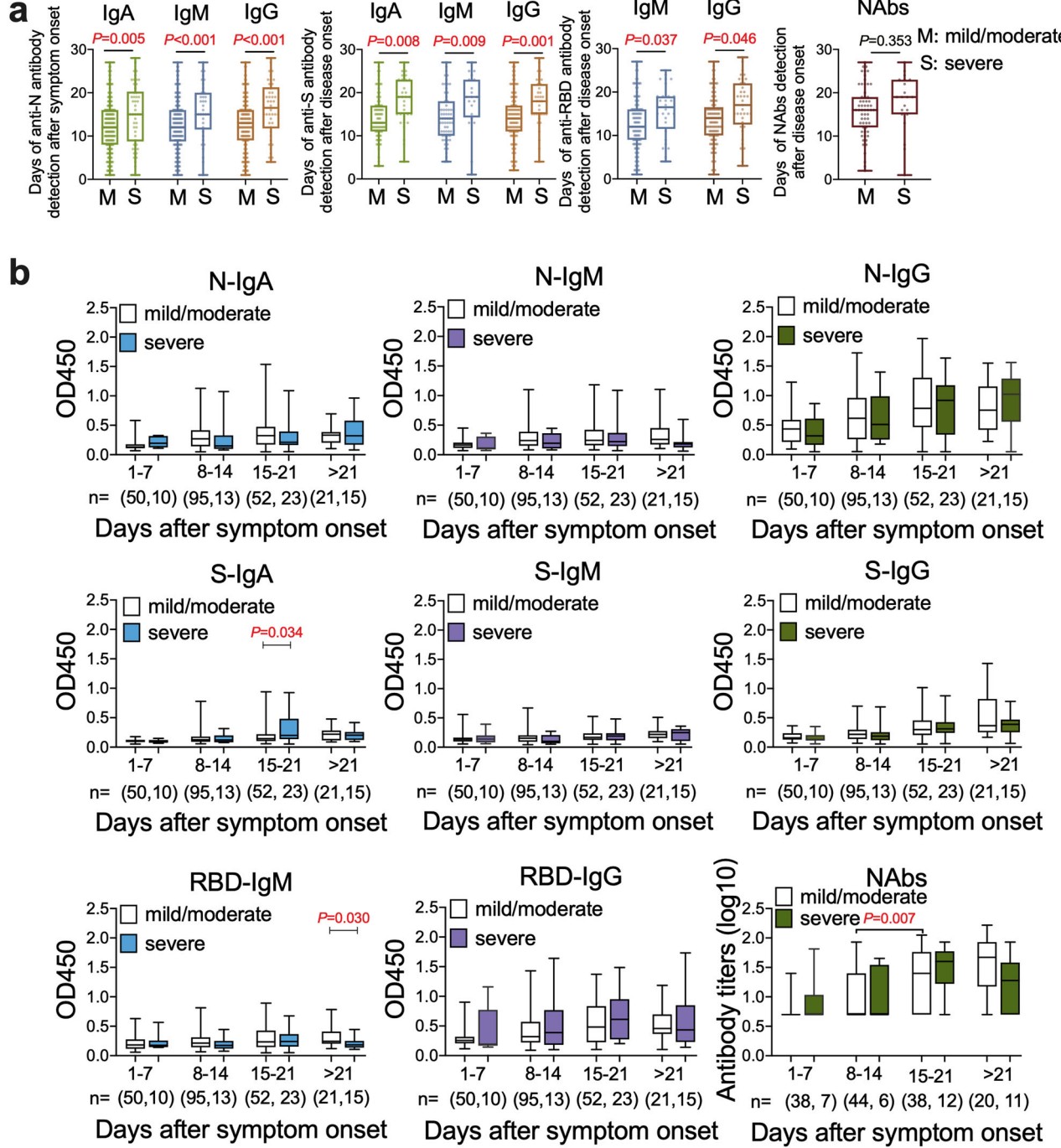

**Fig. 2 The profiles of antibodies in mild/moderate and severe COVID-19 patients. a** The appearance time of IgA, IgM, IgG and total neutralizing antibodies (NAbs) between mild/moderate and severe cases. The onset time of anti-SARS-CoV-2 antibodies against nucleocapsid (N), and spike (S) proteins, anti-receptor binding domain (RBD) IgM and IgG, and NAbs against SARS-CoV-2 in mild/moderate (M) and severe (S) patients is indicated. **b** The optical density (OD) value of IgA, IgM and IgG antibodies against SARS-CoV-2-N, S and RBD are shown while the NAbs titers showed by geometric mean titers over the time course of the disease. The sample number of mild/moderate and severe were provided. The median and quartiles were represented in the boxplots and the bars represent the maximum and minimum values.

production might provide significant predictive value for disease severity.

There are some limitations of this study. The sample size was small, especially in the early and late sampling dates. Further investigations are needed to be performed by involving a large scale of clinical samples. The longitudinal sampling of the same individual would produce a more robust and accurate kinetics of antibody responses. However, it was difficult to follow the patients to collect the serial samples midst the pandemic during

the study period when there was a huge pressure to ensure the best available treatment and quick discharge of patients to accommodate new ones.

In summary, we show that the host mounts a robust humoral response against SARS-CoV-2, which is mediated by antibodies against a wide range of antigens present in the viral structure. We also show the presence of NAbs in these patients. Levels of antibodies, especially IgG, increase over the disease course while a limited increase is observed in IgA and IgM over the disease

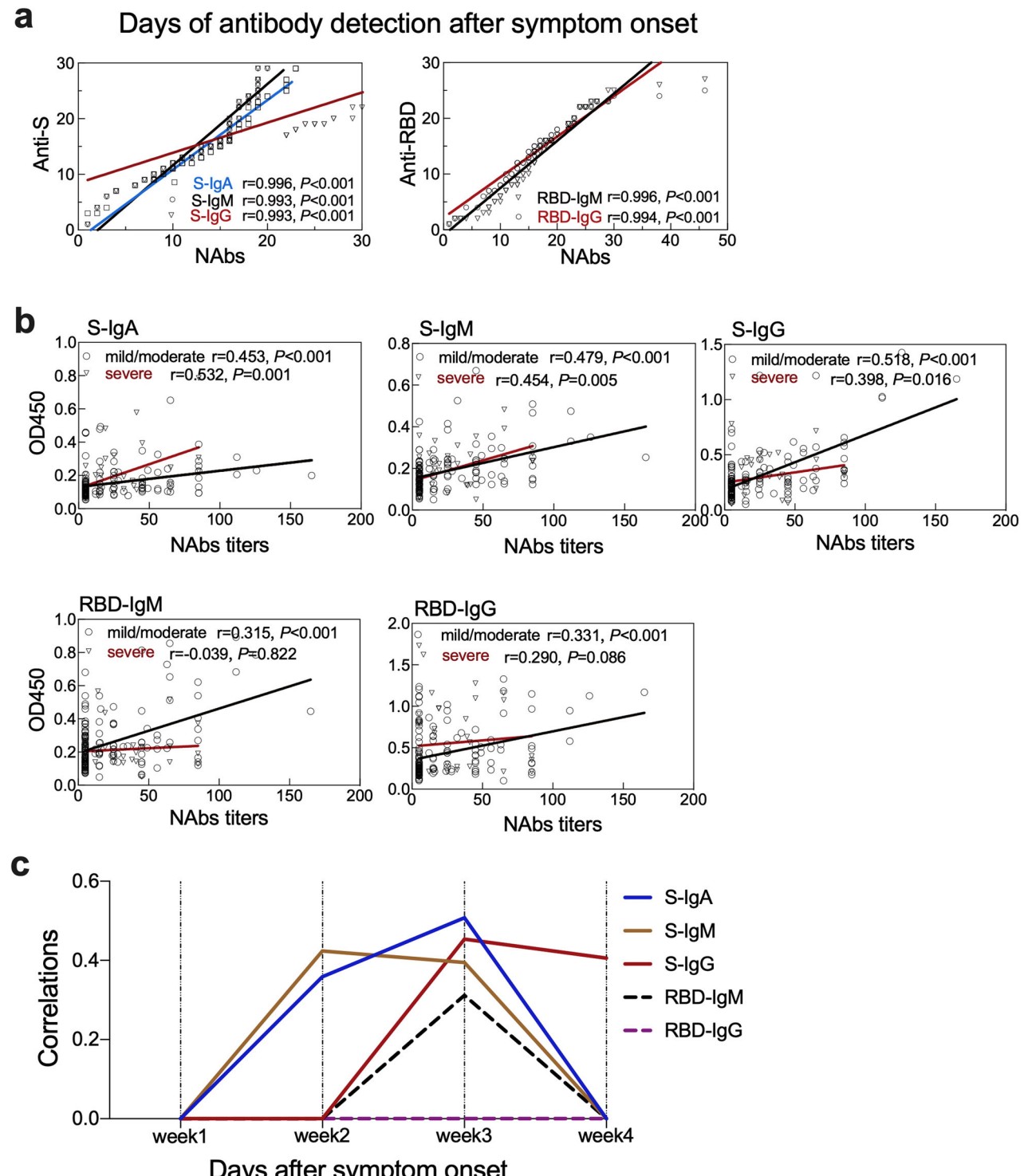

**Fig. 3 Correlations of binding antibodies with neutralizing antibodies (NAbs) in mild/moderate and severe patients. a** The correlations of appearance time between antibodies against spike (S) and receptor-binding domain (RBD) proteins with NAbs. The *x*-and *y*-axis are the number of days when the antibody was detected after symptom onset. **b** Associations of anti-S and anti-RBD antibodies levels expressed as optical density (OD) value with NAbs titers in the mild/moderate and severe patients. **c** The correlations of NAbs titers with the antibodies against S and RBD proteins during the disease course.

course. The delayed onset of antibodies was observed in more severe cases, but at the same time, similar titers were observed in severe cases compared to those with mild/moderate disease. These data indicate that delayed onset of antibodies may contribute to the disease severity, however, more detailed studies are warranted to describe the clear relationship between antibody responses and disease outcome.

## Methods

**Patients and plasma specimens**. In this study, a total of 279 plasma samples were collected from 176 COVID-19 patients from three cohorts, including 218 samples from 140 mild/moderate patients and 61 samples from 36 severe patients. The plasma specimens were collected by using an Ethylene Diamine Tetraacetic Acid anticoagulant tube. In the first cohort, we recruited a total of 123 inpatients (31 severe and 92 mild/moderate cases) from hospitals in Wuhan during the early phase of the pandemic in January 2020. A total of 215 blood samples were taken from the patients

(two serial samples from 92 patients with a 4-day interval and one sample from the remaining 31 patients). The second cohort included a total of 28 hospitalized patients recruited from hospitals in Beijing (5 severe and 23 mild/moderate cases) which provided one blood sample from 17 patients and two samples from 11 patients. Samples of most of these two cohorts have been used in a previous study[7]. Additionally, we recruited a third cohort which included a cluster of 25 patients from Harbin in April. One plasma sample was taken from each patient. All the blood samples were collected between 1 and 46 days of the disease onset. The diagnosis was based on the examinations of nucleic acids and lung computed tomography according to the diagnostic guidelines[12]. The plasma specimens were collected from the patients. The demographic data and clinical diagnosis were recorded.

**Enzyme-linked immunosorbent assay (ELISA)**. The presence of antibodies against SARS-CoV-2 N, S, and RBD proteins were examined by indirect enzyme-linked immunosorbent assay (ELISA) established as previously reported[3]. The N protein was expressed by our group as reported previously. The full-length ecto-domain of S protein and RBD protein were produced in 293T cells with a purity of ≥90% (Sino Biological, Beijing, China). The optimized coating concentrations of N, S, and RBD proteins were determined to be 5, 10, and 5 ng per well. The optimal plasma dilution was 1:400. The polyclonal anti-human IgA (α chain specific) antibody from rabbit (Cat. 309-035-011), anti-human IgM antibody (Fc5μ fragment specific) (Cat. 109-035-043) from goat were from Jack Immuno Research Inc (West Grove, PA, USA). The anti-human IgG (Fc specific) antibody produced in goat (Cat. A0170) was from Sigma-Aldrich (St. Louis, MO, USA).

The ELISAs cut-off values were decided by the mean values and standard deviation (S.D.) of healthy plasma samples by calculating the mean absorbance at 450 nm (A450). The cut-off values of IgA, IgM, and IgG were 0.1, 0.13, and 0.3 for N, and 0.14, 0.2, and 0.21 for S protein, respectively. The cut-off values of IgM and IgG for RBD were 0.2 and 0.3, respectively.

**Microneutralization assay**. The plasma samples were diluted in serial two-fold dilutions from 1:10 to 1:320, then mixed with equal volumes of SARS-CoV-2 at a dose of 100 TCID50 (50% tissue culture infective dose) determined in Vero cells (African green monkey kidney epithelial cells, American Type Culture Collection, CCL-81). The TCID50 was used to evaluate the viral titers by calculating the infective dose causing a 50% cytopathic effect. The mixtures were incubated at 37 °C for 1 h, then 100 μl of the mix was added in quadruplicate to a monolayer of Vero cells (ATCC, CCL-81) cultured in 96-well microtiter plates. The virus-plasma mixture was removed after 1 h and 200 μl fresh growth medium was added to each well. The virus back-titration was included in each test. The plates were incubated for 5 days and the cytopathic effect was observed. The NAbs titers were calculated by using the Reed-Muench method and showed as geometric mean titers (GMTs)[13]. The schematic procedure and representative results are shown in the supplementary file (Supplementary Fig. 1).

**Immunofluorescence assay**. Vero cells were fixed in 4% formaldehyde, permeabilized with 0.5% Triton X-100, and incubated with 5% bovine serum albumin (BSA). The anti-SARS-CoV-2 nucleocapsid antibody, prepared by our group was used as the primary antibody and IRDye Fluor800-labeled anti-mouse IgG (Cat. 926-32210, Li-Cor, USA) was used as a secondary antibody. The nuclei were stained with 4′,6-diamidino-2-phenylindole (DAPI; Sigma, St. Louis, MO, USA). The fluorescence intensity in the cells was scanned by using an Operetta high-content imaging system (Perkin Elmer, Waltham, MA, USA).

**Quantitative real-time PCR**. The nucleic acids were extracted by using TRIzol (Thermo Fisher Scientific, Waltham, MA, USA) according to the manufacturer's instructions. Reverse transcription real-time PCR was performed using the AgPath-ID One-step RT-PCR Kit (Thermo Fisher Scientific). The expression levels of SARS-CoV-2 N RNA in cultured cells were determined according to a standard curve line. Primers targeted to the SARS-CoV-2 N gene were as described in the previous report[1].

**Ethics approval**. The protocol of this study was approved by the Ethical Review Board of Wuhan Jinyintan Hospital, Infectious Disease Hospital of Heilongjiang Province at Harbin, and Institute of Pathogen Biology, Chinese Academy of Medical Sciences & Peking Union Medical College. Written informed consent was obtained from each healthy volunteer, patients suffering from common respiratory tract infections and COVID-19 patients from the hospital in the city of Harbin before enrollment. For the COVID-19 patients from the hospitals in Wuhan and Beijing during January 2020, the written informed consent was waived in the light of this emerging infectious disease of high public health relevance, which was approved by the Institutional Ethical Review Board.

**Statistics and reproducibility**. The details of the experimental designs performed in this study are given in the respective sections of methods. The categorical variables were described as frequency rates and percentages, and continuous variables were described using the median with interquartile range (IQR). Two-tailed unpaired $t$-test was used for group comparison of continuous variables, and $\chi^2$ test was used for group comparison of proportional categorical variables. Correlation analysis was evaluated by the Spearman's rank correlation test. Two-sided $P < 0.05$ was considered statistically significant. All statistical analyses were conducted using SPSS version 19.0 and R version 3.6.1.

**Reporting summary**. Further information on research design is available in the Nature Research Reporting Summary linked to this article.

## Data availability
The source data underlying the graphs and charts in the main and Supplementary figures are available in Supplementary Data 2.

## Code availability
Details of publicly available software used in the study are given in the "Methods". No other custom code or mathematical algorithm that is deemed central to the conclusions was used.

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

## Acknowledgements
We would like to thank the clinicians who contributed to sample collection and transportation. This study was funded in part by the National Major Science & Technology Project for Control and Prevention of Major Infectious Diseases in China (2017ZX10103004, 2017ZX10204401, 2018ZX10734404), the Chinese Academy of Medical Sciences (CAMS) Innovation Fund for Medical Sciences (2016-I2M-1-014), The Non-profit Central Research Institute Fund of CAMS (2020HY320001, 2019PT310029). The funders had no role in the design and conduct of the study; collection, management, analysis, and interpretation of the data; preparation, review, or approval of the manuscript; and decision to submit the manuscript for publication.

## Author contributions
J.W.W., D.Y.Z., L.S., L.H.W., and L.L.R. conceived and designed experiments. L.G., C.W., Y.Y.W., Y.W., H.C.W., and G.W. performed the experiments. D.Y.Z., S.Y.Y., L.H.W., Y.F.H., H.C., JunW.W., and D.Y.Z. contributed clinical samples and clinical data collection. L.L.Z., L.L.R., D.C., C.D.C., D.Y.Z., and J.W.W. analyzed the data. L.L.R., D.C., L.L.Z., L.S., C.D.C., D.Y.Z., and J.W.W. wrote the manuscript. All authors reviewed the manuscript.

## Competing interests

The authors declare no competing interests.
