## [Peer Review File · Communications Biology]

This manuscript has been previously reviewed at another Nature Research journal. This document only contains reviewer comments and rebuttal letters for versions considered at Communications Biology.

Reviewers' comments:

Reviewer #1 (Remarks to the Author):

In their manuscript Ren and colleagues describe the kinetics of antibody responses to N, S and RBD in COVID-19 patients. The paper is of course interesting but has a few problems that need to be addressed.

Major points

- 1) The data analysis, e.g. in Figure 1A is not only driven by % of positives at any given time point but really by sample numbers. For many time points, especially early and late ones there are only very few samples available (e.g. 2-3) which increases the error. Sampling the same individuals longitudinally would produce much more correct answers and this might also look different from the kinetics presented here.
- 2) There is a disconnect between the data shown in Figure 1A and 1B. The data in 1B is basically only driven by available number of samples per day and should be removed.
- 3) Sensitivity and specificity of the different assays should be defined.
- 4) The assays were only performed at one serum dilution and are not quantitative. This needs to be stated, including in the abstract.
- 5) The analysis in Figure 2A makes no sense as it is driven by sample numbers, not by % reactivity.
- 6) Figure 3A is hard to understand and also makes no sense since, again, it is driven by sample numbers.

Minor points

- 1) Line 74/75: Please update the numbers.
- 2) Line 138-139: These are very low positivity rates. Most studies report >90 seroconversion of anti-IgG. Is the low seroconversion rate here a function of the low sensitivity of the authors' assay?
- 3) Line 158: Please define 'MERS-CoV'.
- 4) Line 161: 'experience with other'
- 5) Line 198: Define 'EDTA'.
- 6) Line 210: 'diagnostic guidelines'
- 7) Line 217: 'in 293T'.
- 8) Line 232: Define "TCID50"
- 9) Line 233: 'in Vero'
- 10) Line 240: 'results are shown'

11) Line 244, 247: Define 'BSA', 'DAPI'

12) Line 251: 'using Trizol'

Response to the reviewers

Title: The Kinetics of Humoral Response and its Relationship with the Disease Severity in COVID-19

MS no.: COMMSBIO-20-1715A

Reviewer #1 (Remarks to the Author):

In their manuscript Ren and colleagues describe the kinetics of antibody responses to N, S and RBD in COVID-19 patients. The paper is of course interesting but has a few problems that need to be addressed.

Response: We really want to thank the reviewer for finding our study interesting. We really appreciate your feedback. The point by point response to your comments are given below.

Major points

1.1 The data analysis, e.g. in Figure 1A is not only driven by % of positives at any given time point but really by sample numbers. For many time points, especially early and late ones there are only very few samples available (e.g. 2-3) which increases the error. Sampling the same individuals longitudinally would produce much more correct answers and this might also look different from the kinetics presented here.

Response: We really want to thank the reviewer for this insightful comment. We agree with the reviewer and were aware of this limitation. That's why we indicated the sample numbers below the figure to ensure transparency and make the readers aware of this limitation at early and late time points. As for the sample size, especially in the early and late sampling dates, we analyzed the positive rate in simulation by using cubic equation, which will partially correct the bias caused by the sample size. We do agree

with the reviewer that sampling the same individuals longitudinally would produce much more robust data. However, it is hard to follow the patients to collect the serial samples midst the pandemic during the study period when there was a huge pressure to ensure the best available treatment and quick discharge of patients to accommodate new ones. However, to further clarify and ensure that readers are aware of this limitation, we have added this limitation in Discussion section in our revised manuscript. Please refer to lines 184 to 191.

1.2 There is a disconnect between the data shown in Figure 1A and 1B. The data in 1B is basically only driven by available number of samples per day and should be removed.

Response: We have removed the figure 1B from the original version according to the suggestion. Please refer to lines 144 to 145 in the revised manuscript.

1.3 Sensitivity and specificity of the different assays should be defined.

Response: The specificity of the nucleocapsid (N) protein has been published (Guo et al. *Clin Infect Dis.* 2020. Cia330) and we have included the reference in the revised manuscript. We did not put those data here to avoid duplication. We showed the cross-reactivity of SARS-CoV-2-N protein with SARS-CoV-N but not with the N protein of other human coronaviruses-OC43, 229E, HKU1, NL63 and MERS-CoV. The full-length of spike (S) and receptor binding domain RBD were commonly used proteins to evaluate the antibodies against SARS-COV-2. We noticed that there were cross-reactivities of S protein between the SARS-CoV-2 and human coronavirus OC43. However, the cross-reaction should have no significant influence on the results as the cut-off was decided according to the OD450 value in healthy people and COVID-19 patients. Till now, there are no “gold” standards to decide the sensitivities of ELISA assays for antibody detection. We have described the concentration of

coating proteins, the dilutions of antibodies and samples, which were critical to distinguish the positive plasma from the negative ones.

1.4 The assays were only performed at one serum dilution and are not quantitative. This needs to be stated, including in the abstract.

Response: We thank the reviewer to point out this issue. The antibody titers are presented as OD values as no other standards are available. The dilution used was identified by using our control experiments that were used to optimize the appropriate serum dilution. The level of neutralizing antibodies was detected as GMT, a well-established method for novel antibodies. We have added this in the Abstract section in our revised manuscript. Please refer to the revised abstract.

1.5 The analysis in Figure 2A makes no sense as it is driven by sample numbers, not by % reactivity.

Response: We do agree with the reviewer that the sample size definitely influenced the detection rate of antibodies. However, as the sampling acquisition was random, the results would reflect the actual situation of the studied patients. We have included the case number in the mild/moderate and severe cases group in the text. Please refer to lines 104 to 105 in the text.

1.6 Figure 3A is hard to understand and also makes no sense since, again, it is driven by sample numbers.

Response: This figure shows the correlation between the appearance time of binding antibodies and neutralizing antibodies in COVID-19 patients. The results indicated that the neutralizing antibodies were related to the antibodies against S or RBD. Such findings are important to assess the antibody patterns that are related to

overall neutralizing ability of the antibodies. The analysis is based on the total detected antibodies of all the samples, but not depended on the daily sample size. In addition, 279 blood samples from 176 patients, that were used to derive these graphs should be sufficient to meet the statistical sample size requirement.

Minor points

1.7 Line 74/75: Please update the numbers.

Response: Thank you for pointing out this issue. We have updated the numbers. Please refer to lines 75 and 76 in the revised manuscript.

1.8 Line 138-139: These are very low positivity rates. Most studies report >90 seroconversion of anti-IgG. Is the low seroconversion rate here a function of the low sensitivity of the authors' assay?

Response: The positive rate of IgG was based on S protein as antigen. The positive rate was comparable to other report. In our study, the positive rate of anti-S IgG in convalescent patients was 81.1% (90/111). The results were comparable to other published reports. For instance, in the study by Robbiani DF et al, the positive rate of anti-S IgG was 70% in convalescent individuals. The presence of patients in the acute phase will further reduce the overall antibody positive rate. {Robbiani DF, Gaebler C, Muecksch F, et al. Convergent antibody responses to SARS-CoV-2 in convalescent individuals [J]. Nature, 2020. doi:10.1038/s41586-020-2456-9}.

1.9 Line 158: Please define 'MERS-CoV'.

Response: We have added the full name of MERS-CoV, that is “Middle East

Respiratory Syndrome Coronavirus”, which is also a coronavirus. Please refer to lines 158 to 159.

1.10 Line 161: ‘experience with other’

Response: We have revised the sentence. Please refer to line 162 in the revised manuscript.

1.11 Line 198: Define ‘EDTA’.

Response: We have provided the expansion of EDTA as “Ethylene Diamine Tetraacetic Acid” Please refer to line 208.

1.12 Line 210: ‘diagnostic guidelines’

Response: We have revised it. Please refer to line 220.

1.13 Line 217: ‘in 293T’.

Response: We have revised it. Please refer to line 228.

1.14 Line 232: Define ‘TCID50’

Response: TCID50 is the “50% tissue culture infective dose”. The TCID50 was used to evaluate the viral titers by calculating the infective dose causing 50% cytopathic effect. In this study, we use TCID50 to determine the dilution of plasma which will protect 50% of cultured cells from cytopathic effect caused by viral infections. Please refer to lines 243 to 245.

1.15 Line 233: 'in Vero'

Response: We have revised it. Please refer to line 244.

1.16 Line 240: 'results are shown'

Response: We have revised it. Please refer to lines 252 to 253.

1.17 Line 244, 247: Define 'BSA', 'DAPI'

Response: We added the expansion of BSA (bovine serum albumin) and DAPI (4',6-diamidino-2-phenylindol). Please refer to lines 256 and 259-260.

1.18 Line 251: 'using Trizol'

Response: We have revised it. Please refer to lines 264 to 265.